# Apolipoprotein E, a Crucial Cellular Protein in the Lifecycle of Hepatitis Viruses

**DOI:** 10.3390/ijms23073676

**Published:** 2022-03-27

**Authors:** Yannick Tréguier, Anne Bull-Maurer, Philippe Roingeard

**Affiliations:** 1INSERM U1259 MAVIVH, Université de Tours et CHU de Tours, 37032 Tours, France; yannick.treguier@univ-tours.fr (Y.T.); anne.bull@univ-tours.fr (A.B.-M.); 2Plateforme IBiSA des Microscopies, Université de Tours et CHU de Tours, 37032 Tours, France

**Keywords:** viral hepatitis, hepatitis C virus, hepatitis B virus, apolipoprotein E, lipid metabolism

## Abstract

Apolipoprotein E (ApoE) is a multifunctional protein expressed in several tissues, including those of the liver. This lipoprotein component is responsible for maintaining lipid content homeostasis at the plasma and tissue levels by transporting lipids between the liver and peripheral tissues. The ability of ApoE to interact with host-cell surface receptors and its involvement in several cellular pathways raised questions about the hijacking of ApoE by hepatotropic viruses. Hepatitis C virus (HCV) was the first hepatitis virus reported to be dependent on ApoE for the completion of its lifecycle, with ApoE being part of the viral particle, mediating its entry into host cells and contributing to viral morphogenesis. Recent studies of the hepatitis B virus (HBV) lifecycle have revealed that this virus and its subviral envelope particles also incorporate ApoE. ApoE favors HBV entry and is crucial for the morphogenesis of infectious particles, through its interaction with HBV envelope glycoproteins. This review summarizes the data highlighting the crucial role of ApoE in the lifecycles of HBV and HCV and discusses its potential role in the lifecycle of other hepatotropic viruses.

## 1. Apolipoprotein E

The apolipoproteins are components of the lipoproteins and are classified in multiple classes. Among them, the apolipoproteins A, C and E belong to the human exchangeable apolipoproteins class which, as they are encoded by genes belonging to a multigene family, have the same genomic structure [1,2,3]. The human genetic information allowing their expression are located on three different chromosomes, and *APOE* locus is on the 19th human chromosome. It encodes for a 317 amino-acids (AAs) precursor of the apolipoprotein E (ApoE) containing a 18 AAs N-terminal (N-ter) peptide signal, which is cleaved to obtain a mature ApoE form of 299 AAs. It has six O-glycosylation and two N-glycosylation sites [1,4,5,6,7,8] and is organized into two main domains linked by a protease-sensitive loop of 20 to 30 AAs [5,8,9]. The N-ter domain (AAs 1-191) is a four-helix bundle with a receptor-binding region on the fourth helix that is responsible for the interaction with cellular receptors, such as those of the low-density lipoprotein receptor (LDL-R) family, the very low-density lipoprotein receptor (VLDL-R), scavenger receptor class B type 1 (SR-B1), ApoE receptor-2, and cell membrane heparan sulfate proteoglycans (HSPGs). The C-terminal (C-ter) domain of ApoE contains an amphipathic α-helix responsible for binding to lipoproteins (AAs 244-272) [5,8,9]. The *APOE* locus has three major alleles: ε2, ε3 and ε4 [10]. They encode the ApoE isoforms ApoE2, ApoE3 and ApoE4, respectively, which differ by a single AA substitution in position 112 or 158. ApoE3 (cys112, arg158) is the most common isoform, present in about 79% of the human population worldwide, and is considered to be the normal form of the protein. These isoforms differ considerably in terms of their receptor-binding or functional activities. Indeed, ApoE2 (cys112, cys158) has been reported to be defective for receptor binding and has been implicated in several diseases, including type III hyperlipoproteinemia [11]. ApoE4 (arg112, arg158) has low molecular stability, which influences its ability to bind lipoproteins, and has been implicated in the development of atherosclerosis and neurological disorders. It is an important factor in the onset of Alzheimer’s disease [9,12,13,14].

ApoE is produced principally in the liver, brain, adrenal glands, testes, kidneys and macrophages [9,15,16]. As an exchangeable plasma apolipoprotein, ApoE may circulate freely between lipoproteins, depending on the lipid composition of the various structures, and in an isoform-specific manner. It can be found in a soluble form (a stable lipid-free state) in the bloodstream or associated with lipoproteins in lipid membranes. It is found on the surface of chylomicrons, VLDLs, intermediate-density lipoproteins (IDLs), and a subgroup of high-density lipoproteins (HDLs). The lipid and apolipoprotein composition of lipoproteins determines their size, shape and functions, and strongly influences their biological effects on human health [17]. One major function of ApoE is the regulation of lipid metabolism and transport from the liver to peripheral cells through high-affinity interactions with cell-surface lipoprotein receptors, promoting the clearance, by endocytosis, of plasma lipoproteins, including VLDLs and remnant lipoproteins in particular. It also plays a key role in the morphogenesis of lipoproteins in hepatocytes. High levels of hepatic ApoE lead to increases in triglyceride-rich VLDLs secretion involving the C-ter domain of ApoE. Thus, ApoE participates in lipid content homeostasis at the plasma and tissue levels [9,18]. ApoE also has lipid-unrelated functions associated with exosomes and endosomal intracellular vesicles, and is clearly involved in several stress-response processes influencing mitochondrial function, endoplasmic reticulum (ER) stress and the immune response [19,20].

## 2. Hepatitis C Virus

The hepatitis C virus (HCV) has been extensively described, and much is known about its dependence on lipid metabolism and the actors and associated pathways involved in its lifecycle. The infection caused by this virus was first described in 1975 as a non-A, non-B viral hepatitis (NANBH), as most of the observed cases of transfusion-associated hepatitis had no serological markers for the hepatitis A (HAV) or B (HBV) viruses. The virus was named “HCV” in the late 1980s, following the accumulation of evidence for a new viral hepatitis agent transmitted in blood [21,22,23,24]. About 1.5 million new cases of HCV infection occur each year. About 55 to 85% of acute HCV infections develop into chronic liver infections, accounting for the 58 million chronically infected patients worldwide. Furthermore, 15 to 30% of chronically infected patients go on to develop cirrhosis and/or hepatocellular carcinoma (HCC) in the 20 years following infection; HCV is the leading cause of liver-related death and is now considered as a global health issue (WHO) [24,25,26]. The introduction, since 2011, of combined direct-acting antiviral agents (DAAs) targeting non-structural proteins has revolutionized hepatitis C treatment [27,28]. However, this outstanding success is tempered by the continuing difficulty in eradicating the virus worldwide, as it remains difficult to screen subjects for treatment. The development of a prophylactic vaccine against HCV therefore remains essential [27,28,29].

### 2.1. ApoE on HCV Particles

HCV is an enveloped, single-stranded positive RNA virus from the *Flaviviridae* family. Its genome encodes 10 proteins, including three structural proteins: the core protein that makes up the capsid, and E1/E2, two glycoproteins embedded in the viral envelope. Several components of VLDLs, such as ApoE, B, C1, triglycerides, and cholesterol have been reported to be components of mature HCV viral particles, demonstrating the close relationship of HCV with the lipid metabolism and associated pathways during its lifecycle (Figure 1). These hybrid particles are known as lipo-viro particles (LVPs); they are highly heterogeneous in size, density, and associated structural viral proteins and lipoprotein components [30,31,32,33,34]. Exchangeable apolipoproteins at the surface of LVPs may account for the observed variability of density, as ApoE, B and C1, which interact with circulating triglyceride-rich lipoproteins, facilitating lipid exchanges. As a result, chronic HCV infection has a strong impact on the plasma lipidome. Total cholesterol, LDL, and non-HDL-cholesterol levels in the plasma of HCV-positive patients have been reported to decrease progressively relative to those in healthy controls, regardless of viremia. By contrast, triglyceride and HDL levels are higher in the plasma of HCV-positive patients than in controls [35]. Conversely, Felmlee et al. observed a robust increase in HCV RNA levels in the very low-density fraction of plasma from patients following a high-fat meal, demonstrating a dependence of the density of circulating HCV particles on the lipid composition of the lipoproteins circulating in the plasma [36].

ApoE on circulating particles may interfere with the immune system, providing the virus with partial protection against neutralization, as shown by the differences in the levels of exposure of epitopes B and C of the E2 glycoprotein on the surface of virions between ApoE-expressing HCV-producing culture systems and ApoE-knockout systems. These epitopes could be targeted by various anti-E2 antibodies (Abs) only when ApoE was depleted, suggesting that ApoE may modify the conformation of the HCV envelope proteins, potentially masking several of their immunogenic domains and thereby preventing recognition by neutralizing Abs [37]. Interestingly, one recent study aiming to develop an HCV vaccine based on ApoE-associated envelope proteins showed that, in the presence of ApoE, the HCV envelope proteins had a better conformation, more closely resembling that of the envelope proteins on the surface of virions. The Abs induced by immunization with these ApoE-associated envelope proteins were better able to neutralize the virus, indicating that the HCV envelope protein epitopes exposed better mimicked the epitopes presented by the virions [38]. The interaction between ApoE and HCV envelope proteins may, therefore, be an important element for the development of effective hepatitis C vaccines.

Although the HCV E2 glycoprotein is strongly associated with the viral entry by interacting with CD81, claudin and occludin [39,40,41,42], the discovery that ApoE was present on LVPs justified studies on the role of ApoE in the process by which HCV enters cells (Figure 1). The removal of cell surface-associated HSPGs, the abolition of HSPG binding by ApoE, and heparin treatment clearly inhibited the attachment of HCV to hepatocytes, demonstrating the specific involvement of ApoE in the process of attachment to host cells [43,44,45,46]. LDL-R and SR-B1, which were already known to interact with ApoE, have been identified as specific receptors and cofactors for HCV entry into hepatocytes, in association with the cholesterol transporter Niemann Pick C1-like 1 (NPC1L1) [47,48,49]. Abs targeting ApoE readily neutralize HCV infection in a dose-dependent manner, highlighting ApoE as one of the major elements involved in viral binding to target cell receptors [50]. Moreover, under hypoxia conditions which are more relative to the physiologic conditions in the liver, the induced VLDL-R expression was associated with a most efficient uptake of LDLs and VLDLs through the recognition of lipoproteins-associated ligands like ApoE [51,52]. Thus, when HCV infection experiments were performed under limited oxygen concentration, the HCV entry process was most efficient in this condition. Most importantly, ectopic expression of VLDL-R in cells under normoxia allowed for the obtaining of the same efficiency of infection as cells under hypoxia, highlighting the specific role of VLDL-R in HCV infection [53]. HCV entry through VLDL-R in a hypoxia condition was associated with a recognition of both ApoE and E2 by the receptor and mediated a clathrin-mediated endocytosis [53]. These observations were supported by the description of ApoE ε2, which cannot bind to receptors due to its three-dimensional conformation, resulting in relative protection against the complications of HCV infection [54].

Thus, the ApoE present on infectious HCV particles enables these particles to attach to cells through interactions with HSPGs, and interacts with LDL-R, VLDL-R, SR-B1 and/or NPC1L1 via its N-ter domain for entry into hepatocytes (Figure 1). The three-dimensional conformation of ApoE also clearly modulates the efficiency with which it recognizes receptors.

### 2.2. ApoE and the Intracellular Lifecycle of HCV

Following its entry into host cells via a clathrin-mediated pathway, the viral genome is replicated in specific modified ER vesicles to generate the viral proteins. Most HCV proteins remain associated with the ER membranes [32]. ApoE has been identified as a key factor in HCV morphogenesis, as highlighted by the decreases in intracellular HCV RNA levels and virion secretion observed following the addition of ApoE-targeting siRNA to the HCV cell culture (HCVcc) system [50]. Furthermore, the assembly and release of the virus inhibited by ApoE-specific knockdown in HCV-producing cells can be re-established by the ectopic expression of ApoE in the system [55]. Interestingly, siRNAs targeting ApoB or microsomal triglyceride transfer protein (MTTP), another regulator of lipid metabolism, have no effect on virus production or HCV assembly and release, demonstrating the specificity of the role of ApoE [55]. Further evidence supporting a crucial role for ApoE in the HCV lifecycle is provided by the influence of ApoE on HCV tropism. Hueging et al. demonstrated that ApoB, MTTP, and ApoC1 were dispensable for completion of the HCV lifecycle, but that the simple ectopic expression of ApoE in HeLa cells, which do not naturally support the HCV lifecycle, was sufficient to allow the virus to complete its lifecycle in these cells [56]. Moreover, in Vero cells, the combined expression of ApoE and mir122 also resulted in an efficient HCV-producing system in a cell line not usually permissive for the full HCV infectious cycle [57].

The process of HCV protein production has not been clearly described, as controversy exists concerning the hypothesis relating to the assembly and secretion steps. However, the HCV core protein, located on the cytosolic side of the ER membrane, has been shown to be recruited to diacylglycerol acyltransferase 1 (DGAT1)-generated lipid droplets (LDs) by DGAT1, thereby interlocking the assembly step through RNA recruitment and nucleocapsid formation [58,59]. Various studies have shed light on the specific interaction between HCV proteins and ApoE in the cytosol, at least partially explaining the available data. An interaction of ApoE with the non-structural protein NS5A has also been demonstrated in co-immunoprecipitation and colocalization assays [60]. The C-ter III domain of NS5A has been reported to interact with the core protein via DGAT1 at the interface of LDs [61,62]. Co-immunoprecipitation and colocalization analyses by confocal microscopy have shown that the E1 and E2 glycoproteins interact with ApoE, forming a three-protein complex initiated principally by the ApoE and E2 proteins [31,63]. This interaction occurs in the ER lumen, early in the HCV lifecycle, and is conserved at the surface of the particles [31]. It therefore appears likely that HCV assembly takes place at the point of interaction between LDs and ER, at which the E1/E2-ApoE complexes complete the ApoE-induced RNA-containing capsid, which is then matured in the VLDL maturation pathway for release into the extracellular environment [32,62,64].

All of these elements illustrate the dependence of this virus on ApoE for completion of its lifecycle (Figure 1). This close relationship highlights the adaptation of HCV to the hepatocyte, its major target cell in the host, and the relationships between HCV and the molecules involved in lipid metabolism and associated pathways during the infection process.

## 3. Hepatitis B Virus

Hepatitis B virus (HBV) is a small, enveloped DNA virus from the *Hepadnaviridae* family. Since its discovery in 1966 by H. J. Alter and B. S. Blumberg, a considerable body of knowledge about this virus has accumulated, leading to the development of an effective HBV vaccine and treatments for chronically infected patients based on interferon and nucleoside analogs [65,66,67]. Despite these efficient strategies for fighting infection, HBV remains a global health problem due to the emergence of treatment-specific resistances, a lack of access to the vaccine in various areas, and the persistence of the virus in chronically infected patients [68]. Indeed, this virus, which is transmitted sexually and through blood-to-blood contact, is widely present in both developed and developing countries, with an estimated 296 million chronic carriers worldwide (WHO). These patients, who may or may not be aware of their seropositive status, contribute to the propagation of the virus and are susceptible to major liver complications, such as cirrhosis and HCC [69]. Viral persistence involves the formation of a covalently closed cDNA (cccDNA) in the nucleus of infected cells that cannot be targeted by the treatments currently available. This cccDNA is used as a template for the transcription of viral RNAs, thereby enabling HBV antigens (Ags) to persist in patients [70,71].

### 3.1. Disorders of Global Lipid Metabolism during HBV Infection

Given the hepatic tropism of HBV and the chronic nature of the infection, several studies have considered the potential effect on lipid metabolism and its regulators during the various stages of infection. Several disorders were identified in comparisons of infected and healthy individuals. These disorders included decreases in circulating HDLs and ApoA levels during chronic carriage, and in circulating HDLs, LDLs and ApoB levels during cirrhosis. Moreover, a significant correlation was found between HDLs and viral DNA concentrations in serum [72]. Several studies have tried to unravel the association between HBV status and ApoE concentration and isoforms. Circulating ApoE levels seem to change progressively with the severity of HBV infection. Indeed, regardless of isoform, gradual increases in serum ApoE concentration have been observed in patients with acute hepatitis, cirrhosis, and HBV-induced HCC [73,74]. However, other groups have reported opposite impacts of ApoE isoforms on HBV infection. In northern China, the ApoE2 isoform was found to be associated with the highest risk of HBV infection, whereas an Italian study reported ApoE2 and ApoE4 to be associated with relative protection against HBV [75,76]. Finally, other groups have confirmed an association of ApoE3 with a higher rate of progression of HBV infection to cirrhosis and HCC [73,74]. These findings may reflect this isoform being the most common worldwide, but also the most functional form of the protein.

### 3.2. Presence of ApoE on HBV Particles and Its Role in the Viral Entry Process

The HBV genome has four open reading frames (ORFs), one of which encodes the envelope glycoproteins and generates the PreS1, PreS2 and S sequences (5′ to 3′). Three envelope glycoproteins—HBV L, M and S—are produced from this ORF. HBV L contains all the AAs encoded by the ORF (12 AAs from PreS1, 55 AAs from PreS2 and 256 AAs from S) and HBV M contains only the PreS2 and S domains. HBV S is the smallest envelope glycoprotein, containing only the 256 AAs corresponding to the S sequence. These three envelope glycoproteins are part of the envelope of infectious HBV particles [77]. During HBV infection, these particles, together with subviral envelope particles (SVPs), are secreted by infected host cells. SVPs are secreted in large amounts. These empty particles consist of various amounts of HBV envelope glycoproteins forming spheres (containing HBV S and M) or filaments (containing HBV S, M and L), which are secreted by the constitutive secretory pathway and the multivesicular bodies (MVBs) pathway, respectively [77,78,79]. Like the filaments, infectious particles of HBV hijack the MVBs pathway after HBV S glycosylation in the ER [80], and transport to the ER-Golgi intermediate compartment (ERGIC) [81,82]. The ultracentrifugation of supernatant from HBV-infected cells recently revealed that ApoE, which is secreted in large amounts by host cells, is present in fractions containing HBV core protein, HBV L, and HBV DNA, suggesting that ApoE may be associated with infectious particles of HBV. In addition, the removal of particle-associated envelope glycoproteins by trypsin/Triton X-100 treatment abolished ApoE detection in these fractions. Immunoprecipitation assays with Abs targeting ApoE performed on these fractions efficiently precipitated most HBV particles [83]. Moreover, HBV S and ApoE were recently shown to be colocalized in cells expressing both proteins, and ApoE was detected on the surface of SVPs secreted by these cells, as shown by immunogold labeling and electron microscopy [38]. This finding highlights the association of HBV S protein and ApoE, which seems to occur intracellularly, as ApoE and HBV S were also co-immunoprecipitated from the lysate obtained from these cells [38]. The HBV M and L glycoproteins also contains the S domain. It therefore seems likely that all HBV envelope glycoproteins can interact with ApoE, but further studies are required to confirm this. These data indicate that ApoE is a cellular factor specifically incorporated into infectious HBV particles and SVPs through intracellular interactions with the HBV S protein and, potentially, the M and L proteins (Figure 1).

The role of ApoE in the HBV entry process has recently been deciphered. Indeed, new insights were obtained by Qiao and Luo, who described the ability of Abs targeting ApoE to neutralize HBV entry in a dose-dependent manner [83]. The incubation of HBV particles with anti-ApoE Abs reduced infectivity in primary human hepatocytes and HepG2 cells expressing the HBV-specific receptor (HepG2-NTCP cells). This inhibition led to a 90% decrease in HBV cccDNA, DNA and HBe (another, core-derived, HBV protein secreted into the extracellular environment) levels in infected cells [83]. Furthermore, the use of Abs targeting LDL-R, or of siRNA and CRISPR/Cas9 inducing the knockout of the receptor yielded the same results, highlighting the role of LDL-R as a cofactor in HBV entry and explaining the role of ApoE on the particles in mediating ligand binding to the receptor [84]. HBV S has also been reported to interact with glycosaminoglycans (GAGs), facilitating the attachment of HBV to cells [85,86]. These attachment factors are used by several metabolic and pathogenesis pathways and they interact with ApoE in the homeostasis of lipids and immunity [44,45,46]. ApoE may, therefore, also be involved in the attachment of HBV to host cells by this mechanism.

### 3.3. ApoE and the Intracellular Lifecycle of HBV 

The incorporation of ApoE into HBV infectious particles and the specific co-immunoprecipitation of HBV S and ApoE proteins from cells suggested that ApoE might play an extended role in the intracellular lifecycle of HBV [38,83]. HepG2-NTCP cells in which ApoE expression was silenced by both siRNA and CRISPR/Cas9 techniques were, therefore, infected with HBV. A significant decrease in intracellular cccDNA and HBV core protein levels dependent on silencing efficiency was observed and the amounts of both HBe and HBV DNA in the supernatant were affected. This finding indicates that ApoE expression in the host cell also modulates HBV infection. This specific role of ApoE in HBV infection was supported by the absence of an effect of ApoB silencing by siRNA in the same cell line [83]. In HepAD38 cells (a cell line that constitutively produces HBV virions), silencing by siRNA and ApoE knockout demonstrated that ApoE had no impact on viral replication, as attested by the absence of decrease in core protein and HBV DNA levels. However, HBV DNA levels in the supernatant were affected, due to the decrease in secretion of enveloped HBV particles, but not of non-enveloped nucleocapsid. These data highlight the important role of ApoE in the secretion of infectious particles of HBV, through interactions with HBV S during the intracellular lifecycle of HBV. Importantly, the re-establishment of ApoE expression by transfection in the ApoE-knockout cell lines restored efficient completion of the HBV lifecycle [83].

The specific roles of ApoE in both HBV infectious particle production and entry into hepatocytes remain incompletely understood, but these recent data clearly demonstrate the importance of this cellular host factor for the HBV lifecycle (Figure 1). HBV should therefore be considered a second hepatitis virus for which ApoE plays a crucial role in the lifecycle.

## 4. Other Hepatitis Viruses

HCV and HBV are the most studied hepatotropic viruses due to the chronic nature of the infections they cause and the associated risk of HCC development. However, other hepatitis viruses may also make use of the lipid metabolism and associated pathways in the completion of their lifecycles. The hepatitis delta virus (HDV) occupies a unique position among hepatotropic viruses, as it is a satellite virus of HBV. HDV does not encode any viral envelope proteins, instead borrowing the envelope of HBV. Thus, HDV requires HBV as a helper virus for virion assembly, envelopment and spread in host cells [87,88]. As ApoE interacts intracellularly with HBV S and participates in HBV entry by interacting with cellular receptors, it seems highly likely that it may play a similar role in the entry of HDV, but this remains to be demonstrated.

Both the hepatitis A and E viruses (HAV and HEV) were previously described as “naked viruses”, but have since been reported to acquire a cellular membrane during their secretion, and are now considered to be “quasi-enveloped” viruses [89,90,91,92,93]. Their quasi-envelopes are of exosomal origin and seem to promote viral escape from the immune system and the spread of the virus in the liver [89,90,91,94,95,96]. Further studies are required to explore the potential role of ApoE in the formation of the quasi-envelope of these viruses. Interestingly, a hijacking of the MVBs pathway has been described for both HAV and HEV, as for HBV [78,95,96,97]. Indeed, ORF3, the second structural protein of HEV, has been shown to interact specifically with the TSG101 component of the ESCRT complex to facilitate HEV secretion, and this same protein has been identified as a component of HAV particles [95,97]. ApoE is associated with exosomes and endosomal intracellular vesicles [20], justifying studies of the possible role of ApoE in the lifecycles of both HEV and HAV. Several observations concerning the relationship between HEV infection and ApoE have already been published. An increase in ApoE concentration was described in HEV-infected patients, during the acute phase of infection, in the Indian population [98]. Another two studies tried to find evidence for a role of ApoE in HEV infection and reported that some single-nucleotide polymorphisms (SNPs) in ApoE genes, such as rs7412 C-to-T, or carriage of the ApoE3 and ApoE4 isoforms, could be associated with a lower risk of HEV infection in Chinese Han men and non-Hispanic populations, respectively [99,100].

## 5. Conclusions

ApoE plays a very important role in the infectious cycle of two major hepatitis viruses causing chronic infections leading to severe liver disease. This role has been known for some years for HCV, a virus closely associated with the lipid metabolism that forms LVPs. However, the role of ApoE in the infectious cycle of HBV has only recently been discovered. Further studies will undoubtedly clarify the role of ApoE at different stages of the infectious cycle of HBV. Moreover, this recent discovery suggests that the ability of other hepatotropic viruses to infect liver cells and spread throughout the liver may also be dependent on ApoE, which should encourage further studies of this aspect.

## Figures and Tables

**Figure 1 ijms-23-03676-f001:**
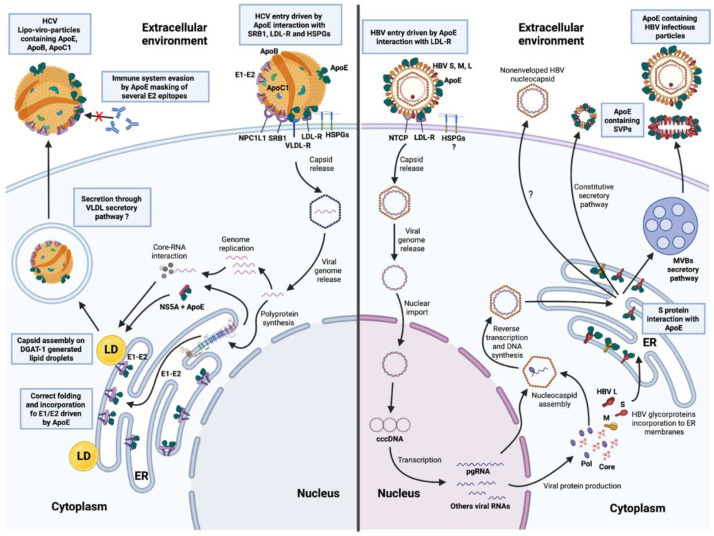
The crucial role of apolipoprotein E (ApoE) in the life cycles of both HCV and HBV. Left panel: HCV lipoviro particle (LVPs, containing lipoprotein components such as ApoE, ApoB, ApoC1 and triglycerides) entry is driven by E2 interaction with CD81, claudin 1 and occludin (not represented in the figure) and ApoE interactions with scavenger receptor class B type 1 (SR-B1), low-density lipoprotein receptor (LDL-R), very low-density lipoprotein receptor (VLDL-R) and heparan sulfate proteoglycans (HSPGs), expressed on the surface of the host cell. After viral entry and capsid release, the viral genome is released into the cytoplasm, where it is replicated and translated into a polyprotein targeted to the endoplasmic reticulum (ER) membrane. E1 and E2 envelope glycoproteins are associated with the ER membrane and interact with ApoE, ensuring their correct folding and future incorporation into the nascent viral particles. At the same time, the non-structural NS5A protein also interacts with ApoE and is targeted to DGAT-1-generated lipid droplets (LD). E1-E2-ApoE complexes interact with the core-RNA complex at an undefined LD-related site. The nascent particles incorporate several lipoprotein components during trafficking and may be secreted via the VLDL secretory pathway. Right panel: ApoE-containing HBV particle entry into host cells is driven by interactions between S protein and the NTCP receptor, but also between ApoE and LDL-R. The HBV capsid is then released, and the HBV genome is translocated to the nucleus to form the covalently closed cDNA (cccDNA). This cccDNA is the template for the production of the pregenomic RNA (pgRNA) and other viral RNAs, which are translated to produce viral proteins. The HBV S, M and L envelope proteins are translocated to the ER membranes, where they interact with ApoE, whereas the core protein interacts with the pgRNA and the Pol protein to form the nucleocapsid. After reverse transcription of the pgRNA by Pol, the HBV DNA-containing neocapsid is then trafficked through the ER for secretion by an unknown pathway, or enveloped by HBV S, M and L to produce ApoE-containing HBV infectious particles that are released into the extracellular environment by the multivesicular bodies (MVBs) pathway. Subviral envelope particles (SVPs) are secreted during the HBV lifecycle: spherical SVPs containing S and M are secreted via the constitutive secretory pathway whereas filamentous SVPs, containing the S, M and L proteins, follow the same secretory pathways as HBV infectious particles. Both types of SVP are thought to incorporate ApoE.

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
