# Peer review of "Apolipoprotein E, a Crucial Cellular Protein in the Lifecycle of Hepatitis Viruses"

_ijms, 2022, doi:10.3390/ijms23073676_

Round 1

Reviewer 1 Report

This manuscript is a well-prepared informative review that summarizes the data highlighting the important role of ApoE in the lifecycles of HBV and HCV and discusses its potential role in the lifecycle of other hepatotropic viruses. More specifically, the manuscript describes the role of ApoE as part of the viral particle, in the viral entry into the host cells and in viral morphogenesis. Furthermore, information is provided on the function of ApoE in mediating evasion from HCV neutralizing antibodies providing the virus with partial protection.

The topic is of high interest. Adequate information is provided for the role of ApoE in HCV infection, which is known for years. Moreover, recent observations concerning the implication of ApoE in HBV and HEV infections are discussed, which brings much novelty and highlights the need for new studies on the interaction of ApoE with HEV and HAV life cycles.

I would suggest some minor changes:

- A comment can be added on the role of ApoE in HCV entry under low oxygen tensions, which are physiologically present in the liver. Βased on previous data, the expression of VLDLR is upregulated under hypoxia (not expressed under atmospheric conditions) and is involved in mediating HCV entry into hepatocytes, and this interaction is mediated by HCV E2 and ApoE (doi: 10.1073/pnas.1506524113). More specifically, the expression of VLDLR is upregulated under hypoxic conditions (doi: 10.1073/pnas.1506524113). As a result, the uptake of LDLs and VLDLs is enhanced (doi: 10.1042/BJ20111377), possibly through the recognition of ligands (such as apolipoprotein) that associate with the lipoproteins (doi: 10.1194/jlr.M500114-JLR200).

- At the end of the first paragraph of the section Hepatitis C virus (lines 79-80), a part of the sentence initiating “The development of a prophylactic vaccine …” is missing.

- Some extra spaces should be deleted: a) in line 119, before the word “HSPG”, b) in line 155 before “by DGAT1”, c) in line 167 before “capsid”, d) in line 331 before “populations”, e.t.c.

Reviewer 2 Report

While the role of ApoE in the infectious cycle of HCV is well established, its role in HBV infection has only recently been demonstrated. It is possible that the ability of other hepatotropic viruses to infect livers may depend on ApoE. My concerns should make the text clearer to a wider readership.

Lines 23-43: Please provide a concise introduction on apolipoproteins, including their genetics, polymorphism, isoforms and phylogeny with a special emphasis on ApoE. It would be helpful to add a new figure showing these features in a graphical form.

It is difficult to integrate some parts of text without more general introduction, e.g.: Figure 1 legend (lines 256-265); metabolic importance of ApoE in HBV-infected individuals (lines 197-208), etc.

Line 40: please indicate a reference for type III hyperlipoproteinemia.

Lines 63-80, references [17-23]: please provide some more recent references.

Lines 82-87: please refer to figure 1 in text and show constituents of the lipo-viro particle in the figure 1.

Lines 117-132: please refer to figure 1 in text when describing HCV particle cell entry.

Please discuss importance of E2 glycoprotein interaction with CD81/claudin/occluding in HCV cell entry and virus tropism in comparison to ApoE interaction with LDL-R/SR-B1/NPC1L1. Seminal papers show importance of E1-E2 glycoproteins in HCV cell entry mechanism and tropism.

Figure 1, lines 256-265: Presence of ApoE, ApoB and ApoC1 on HCV particle is shown without any comment concerning their importance. Also, presence of E1-E2 on surface of HCV particle is shown without any comment concerning their importance in the figure legend. Importance of interactions between HBs protein and the NTCP receptor are correctly described (lines 266-268), in the right panel.

Figure 1, lines 136-149: please show more clearly cytosolic or luminal association of ApoE with HCV core (namely NS5A) and HCV E1-E2 glycoprotein, and HBs in ER. It is not clear in Figure 1, neither in the text.
